# Metabolite Analysis of *Alternaria* Mycotoxins by LC-MS/MS and Multiple Tools

**DOI:** 10.3390/molecules28073258

**Published:** 2023-04-06

**Authors:** Yanli You, Qinghua Hu, Nan Liu, Cuiju Xu, Sunan Lu, Tongcheng Xu, Xin Mao

**Affiliations:** 1College of Life Science, Yantai University, Yantai 264005, China; 2Institute of Food & Nutrition Science and Technology, Shandong Academy of Agricultural Sciences, Jinan 250100, China

**Keywords:** *Alternaria*, modified mycotoxins, metabolic pathway, molecular networking, feature-based molecular networking

## Abstract

*Alternaria* fungi are widely distributed plant pathogens that invade crop products, causing significant economic damage. In addition, toxic secondary metabolites produced by the fungi can also endanger consumers. Many of these secondary metabolites are chemically characterized as mycotoxins. In this study, Q Exactive Orbitrap mass spectrometry was used for the non-targeted analysis of the metabolome of seven *Alternaria* isolates cultured on Potato Carrot Agar (PCA), Potato Dextrose Agar (PDA) and Potato Sucrose Agar (PSA) medium. Due to the difficulty of detecting modified toxins, an analytical strategy with multiple visual analysis tools was also used to determine the presence of sulfate conjugated toxins, as well as to visualize the molecular network of *Alternaria* toxins. The results show that PSA medium exhibits more advantageous properties for the culture of *Alternaria*, with more toxigenic species and quantities and more obvious metabolic pathways. Based on high-resolution tandem mass spectrometry (MS/MS) data, the mycotoxins and their metabolites were mainly clustered into four groups: alternariol (AOH)/alternariol monomethyl ether (AME)/altenusin (ALU)/altenuene (ALT)/dehydroaltenusin (DHA)/Desmethyldehydroaltenusin (DMDA) families, Altertoxin-I (ATX-I) family, tentoxin (TEN) family and tenuazonic acid (TeA) family. Moreover, the PSA medium is more suitable for the accumulation of AOH, AME, ALU, ALT, DHA and DMDA, while the PDA medium is more suitable for the accumulation of ATX-I, TEN and TeA. This research may provide theoretical support for the metabolomics study of *Alternaria*.

## 1. Introduction

*Alternaria* is an omnipresent fungal genus that includes saprophytic, endophytic and pathogenic fungi. It is related to a wide variety of substrates including seeds, plants, agricultural products, animals, soil and the atmosphere [1]. *Alternaria* fungi are a class of important plant pathogens which can cause diseases of crops, cash crops and fruit trees, resulting in serious economic losses [2,3]. In addition to agricultural product losses, *Alternaria* fungi endanger consumers by the production of mycotoxins and secondary metabolites, such as causing vomiting in animals and causing psoriasis in humans [4]. Since the early 1980s, numerous reviews on *Alternaria* mycotoxins have been published in recent decades [5,6,7,8,9]. 

To date, more than 300 species of *Alternaria* were reported globally, and facultative parasitism on plants accounts for more than 95% [10]. At least 70 metabolites from *Alternaria* fungi have been reported in the past. They can be divided into several categories, which include nitrogenous compounds, steroids, terpenoids, pyranones, quinones and phenols. Several metabolites are unique to one *Alternaria* species, but most metabolites are produced by more than one species [8]. The main *Alternaria* mycotoxins belong to the chemical group dibenzo-α-pyrones, which includes alternariol (AOH); alternariol monomethyl ether (AME); altenuene (ALT); cyclotetrapeptides, represented by tentoxin (TEN); and the perylene quinone derivatives altertoxins-I. These mycotoxins are the most frequently studied metabolites produced by *Alternaria* strains on different substrates (tomato [11], wheat [12], cherry [13], pear [14], etc.), and some major *Alternaria* compounds are considered to have potential risks to human and animal health due to their known toxicity and frequent presence as natural contaminants in foods [15]. In addition, mycotoxins may undergo chemical reactions such as oxidation, reduction and hydrolysis or bind to amino acids, glucose, sulfate groups and glutathione in plants and other organisms (e.g., fungi, bacteria, mammals). This can alter the chemical structure of mycotoxins as part of their protection against foreign organisms, thereby further increasing the range of contaminants that may occur [16]. As the sulfate conjugated form of AOH and AME were reported in cereal and cereal-based foodstuffs [17,18], people are more concerned about such modified mycotoxins [19]. There is speculation that conjugated *Alternaria* secondary metabolites, along with their parent forms, could occur in affected foods. Aly et al. found that mouse lymphoma cells were screened for cytotoxic activity in vitro with AOH-SO_3_, which had lower bioactivity than free AOH, but higher bioactivity than free AME [20]. However, there are still many doubts about whether other conjugated forms of *Alternaria* mycotoxins are toxic to animal cells, so the detection of other conjugated mycotoxins is essential to determine a safe level for human and animal consumption.

In order to make a reliable risk assessment, it is necessary to study the metabolic capacity of *Alternaria* fungi and to correctly identify the various mycotoxins. Liquid chromatography-mass spectrometry has become the preferred technology for the accurate analysis of many mycotoxins [21], but due to the lack of commercial mycotoxin standards, modified mycotoxins are often overlooked in routine detection. A visible post-data process procedure with MZmine 2 [22], vender software (Xcalibur) and Global Natural Products Social Molecular Networking (GNPS) [23] was used for the efficient analysis of high-resolution mass spectrometry. Conjugated products were screened with an optimized diagnostic fragmentation filtering module in MZmine 2 and further confirmed with Xcalibur by comparison with commercial standards for unconjugated products. MS/MS spectral data were processed and used to establish a feature based on a molecular networking map in Global Natural Products Social Molecular Networking (GNPS; https://gnps.ucsd.edu (accessed on 15 September 2021)) [24,25] for visualization of *Alternaria* natural product families. A new solution for the detection of modified mycotoxins and the study of natural chemical products was provided [26]. Toxins, as a secondary metabolite of fungi of the *Alternaria* genus, not only have non-negligible hazards, but also have certain biocontrol potential [27,28]. For example, TeA produced by *Alternaria alternata* can effectively inhibit the growth and reproduction of *Macrosiphum rosivorum* [29]. Therefore, many *Alternaria* mycotoxins have the potential to develop into biocontrol agents for many plant diseases. It is very important to screen strains with strong toxigenic ability as biocontrol fungi. The toxigenic ability of the toxigenic strains of *Alternaria* is greatly affected by external physical factors, especially the selection of culture medium. Potato Dextrose Agar (PDA) medium is mostly used as a culture medium when studying the culture of toxigenic conditions. Zeeck and colleagues have proposed a method to release nature’s chemical diversity named the OSMAC (One Strain—Many Compounds) approach, and it reflected that changing some cultivation parameters (such as medium composition, cultivation conditions, etc.) can alter the metabolic characteristics of various microorganisms. Based on the OSMAC approach, different media represent different environments that the fungi are exposed to, requiring them to alter their behavior and produce different metabolomes [30]. Therefore, based on the previous studies of *Alternaria* in PDA solid medium, we re-cultured them in PDA, PCA and PSA medium to comprehensively analyze the production of natural products of *Alternaria*.

In this study, we explored the toxigenic situation of *Alternaria* fungi on three different mediums. The natural products of *Alternaria* produced in the culture medium were analyzed to determine the production rule and metabolic pathway of *Alternaria* mycotoxins, providing a basis for the further study of *Alternaria* mycotoxins. The research methods adopted in this study are shown in Figure 1.

## 2. Results

### 2.1. Identification of Free Metabolites

In this study, seven strains of *Alternaria* fungi were inoculated onto PCA, PDA and PSA media for culture, and metabolites were extracted regularly. In our preliminary examination, target metabolite analysis of the strain samples was performed using ultra performance liquid chromatography-high resolution tandem mass spectrometry (LC-MS/MS) in both negative and positive ionization modes. The results showed that the metabolites had a better response in positive ionization mode than in negative ionization mode (Appendix A). Therefore, we used the positive ionization mode for the analysis of *Alternaria* metabolites. 

By comparison with commercial standards, seven free *Alternaria* species were detected in three mediums, including AOH, AME, ALT, TeA, ALU, TEN and ATX-I (Appendix A). Unfortunately, we did not detect MPA in samples. As a result of limited reference compounds for natural *Alternaria* products, by referring to the published literature and speculating with the relevant compounds [26], we confirmed the presence of two metabolites, dehydroaltenusin and desmethyldehydroaltenusin.

### 2.2. Identification of Sulfated Conjugated Metabolites

High performance liquid chromatography-high resolution tandem mass spectrometry (LC-MS/MS) in positive ionization mode was used to screen the metabolites of *Alternaria* in three culture media samples. For *Alternaria* mycotoxin conjugated metabolites, it was difficult to identify due to the lack of commercial standard mycotoxins. Metabolites are usually a mixture of structurally similar compounds. Due to the common structural features, many compounds in the same class undergo similar MS/MS fragmentations and have several identical product ions and/or neutral losses. Several metabolites displayed a neutral loss of 79.9568 Da by dissociating of SO_3_ in mass spectrum, indicating that sulfate conjugation might be a major modified form of *Alternaria* mycotoxins. For full investigation of the extent of sulfate *Alternaria*, the neutral loss filtering (NLF) function in diagnostic fragmentation filtering (DFF) was performed with MZmine for complete data set monitoring and the neutral loss of all products was plotted [31]. The *m*/*z* was set to 79.9568 for the diagnostic production. Processing the data to generate the neutral loss plot of *Alternaria* (Figure 2), the neutral loss plot could visually extract showing precursor metabolite *m*/*z* and neutral mass losses. Taking the extraction results of strain NO. 196 as an example, a total of six compounds were detected with a neutral loss of 79.9568 (±5 ppm) in medium, including one for PCA, two for PDA and four for PSA plates (Appendix A).

The neutral loss plot of an *Alternaria* spp. extract shows the *m*/*z* and neutral mass losses of the precursor metabolite. The neutral mass loss of SO_3_ (79.9568 Da) is highlighted by a dashed line, with each ‘x’ along the line corresponding to a sulfate conjugated metabolite.

To further validate the compounds and confirm the identity of these screened compounds results, the MS/MS spectra of the free metabolites were compared with the MS/MS spectra of their respective conjugated metabolites using the Xcalibur software. As shown in Figure 3B, conjugated mycotoxin metabolites can yield free mycotoxins with the neutral loss of SO_3_ in MS/MS, and the fragment ions and relative intensities of the conjugated and free ions are the same. A mass shift of 79.9568 and the relative abundance of the isotopic peaks were used to aid in the elemental formula. The ^34^S isotope peak has a mass shift at 1.995 Da above the monoisotopic peak and takes a natural abundance of 4.21%, while the ^13^C*2 isotope peak has a mass shift at 2.005 Da (Figure 3). For sulfated conjugations, there is one S element in the structure. Mass shifts with 1.995 for ^34^S and 2.005 for ^13^C can be used for the confirmation of sulfated mycotoxins. In this way, the complete dataset was confirmed and the detected sulfated conjugates were AOH sulfated, AME sulfated, DHA sulfated and DMDA sulfated.

### 2.3. Analysis of the Production of Toxin by Alternaria

The custom mycotoxin database was established based on identified mycotoxins that contained accurate mass, retention time, compound name and molecular formula of the mycotoxins (Appendix A) for qualitative and quantitative analysis of free *Alternaria* mycotoxins and sulfated conjugates. The existing *Alternaria* toxin standard was used as the quantitative standard to quantify all the detected *Alternaria* toxins in the extracted metabolite samples. In contrast to the free *Alternaria* mycotoxins, accurate concentrations of sulfated conjugates produced by *Alternaria* strains could not be obtained due to the lack of commercial standards. Xcabliur was used to quantitatively analyze the extracts of seven strains of *Alternaria* at different culture times in different media. According to the peak area response of *Alternaria* toxins and hidden toxins in high resolution mass spectrometry, the quantitative analysis of various *Alternaria* toxins was carried out. The quantitative results were used to establish a heatmap using the R package (Heatmap). Heatmaps were for the visualization of quantification analysis to gain insight into features that behave similarly over time, strains and medium type. 

All seven isolated strains were cultured at different mediums of PCA, PDA and PSA at culture times of 10, 20 and 30 days. In the figure, blue represents no toxin detected or a low concentration of toxin detected, while pink represents a detected toxin. The darker the color, the higher the concentration. In this way, we can obtain a relative MS response result, which can also indicate the natural production tendency from isolated fungi. As can be seen from Figure 4, all strains can produce *Alternaria* mycotoxins. The detection rate of *Alternaria* mycotoxins produced by fungi on PSA medium was the highest (72.5%), followed by PDA medium (55.7%), while PCA medium (19.8%) was the lowest. Fungi cultured on PSA medium produced the most abundant amount of *Alternaria* mycotoxins, and then PDA medium. Fungi cultured on PCA medium can barely produce *Alternaria* mycotoxin products, especially sulfated conjugate toxins. Compared with PDA medium, PSA is more advantageous for *Alternaria* culture. It is speculated that it may be related to the different types of carbon sources in the culture medium. Glucose promotes the growth of hyphae, but sucrose is more conducive to the production of spores. Therefore, sucrose in PSA is more conducive to the production of toxins than glucose in PDA.

### 2.4. FBMN-Based Metabolic Pathway Analysis

High resolution tandem mass spectrometry (MS/MS) data collected in positive ionization mode were analyzed with the GNPS molecular networking portal and visualized in Cytoscape. In a molecular network, related molecules are referred to as a ‘molecular families’, and the nodes represent assigned molecular formulae, which are linked via edges representing simple biochemical reactions such as glycosylation, alkylation and oxidation/reduction [24]. The network established during the annotation process is shown in Appendix A). This annotation can be propagated to identify new compounds by converting the mass differences of the edges into structural motifs. For ease of presentation, we simplified the network, which is shown in Figure 5. Four molecular families of *Alternaria* mycotoxins were produced, including AOH, AME, ALU, ALT, DHA and DMDA families as cluster one, ATX-I family as cluster two, TEN family as cluster three and TeA family as cluster four. 

As shown in Cluster one, (Figure 5A) comprises 28 nodes; they clearly show several subclusters associated with identified *Alternaria* mycotoxin families of AOH (C_14_H_10_O_5_), AME (C_15_H_12_O_5_), ALU (C_15_H_16_O_6_), ALT (C_15_H_16_O_6_), DHA (C_15_H_12_O_6_) and DMDA (C_14_H_10_O_6_). Molecular network results show that mycotoxins and their metabolites are strongly metabolically correlated, which is related to their similar structures. They are connected by nodes, either directly or indirectly, indicating that some simple biochemical reactions occur during mold growth, suggesting that some toxins may be metabolites. In the cluster, there are four nodes directly connected to the AME, one of which has a molecular weight difference of 0.001 and can be identified as its isomer, and the exact mass differences of the O interconnecting molecular formulae are *m*/*z* 289.0705, RT: 8.67 and *m*/*z* 257.0813, RT: 5.29. We can identify the *m*/*z* 353.0326 node as a sulfide AME by using its mass difference of 79.957. There are three nodes directly connected to the AOH, and the node with *m*/*z* 259.0607 is an isomer of it. The mass difference between the AOH and the *m*/*z* 191.0704 node is 67.990, and the spectroscopic matching results between them suggests the presence of C_3_O_2_ loss. Nodes with an *m*/*z* of 339.0161 (RT: 6.66) are sulfated conjugation metabolites of AOH. The elemental compositions of AOH-SO_3_ and AME-SO_3_ are linked via the accurate mass difference of CH_2_. We found that the spectra of conjugated AOH and AME contained fragment ions in mass to their free AOH and free AME by manually querying the spectral fragments of the nodes and performing an artificial alignment (Figure 5B). They share a cosine similarity score of 0.92 and 0.93, indicating a high degree of spectral similarity, which supports this identification [23,32]. We selected the TOP5 spectral intensity as the contrast criterion, where the rods with different color criteria are TOP5 for AOH and AME. This example demonstrates how manual comparison of the MS/MS spectra of the molecular families that make up conjugated compounds can also be critical for structural annotation. Furthermore, this example also validates the results of 3.2, again confirming the presence of the sulfate conjugate. The only node connected to the ALU is the metabolite through its loss of H_2_O, and the node of ALT also forms different metabolites through dehydration reactions. The node directly linked to DMDA can be identified as its isomer, which is linked to the isomer of DHA via the accurate mass difference of CH_2_. 

As shown in Figure 5C, cluster two comprises 33 nodes with the identified mycotoxin of ATX-I (C_20_H_16_O_6_). Unfortunately, there are only two nodes directly connected to ATX-I. The mass difference between the node with *m*/*z* 335.0903 and the node with *m*/*z* 317.0809 and ATX-I is 18.012 and 36.021, which is speculated to be H_2_O loss. In addition to the confirmed mycotoxins, the experimental spectrum of toxin cluster two was matched to kynurenic acid (m/z 190.0493, RT: 4.31, C_10_H_7_NO_3_) in the GNPS spectral library. In comparison to the cluster above, TEN (Figure 5D) and TeA (Figure 5E) were small molecular families in molecular networking. Cluster three comprises seven nodes with the identified mycotoxin of TEN (C_22_H_30_N_4_O_4_). The node with *m*/*z* 829.4588 (RT: 5.94) was a dimer of TEN. The nodes with *m*/*z* 429.2502 (RT: 6.30) and *m*/*z* 443.2759 (RT: 6.94) were the products of TEN methylated. Cluster four comprises only three nodes with the identified mycotoxin of TeA (C_10_H_15_NO_3_). The two nodes with *m*/*z* 239.1316 and *m*/*z* 239.1429 are the precursor ions of TeA. Further studies are needed to investigate why the sulfated conjugations products of DHA (Figure 5F) and DMDA (Figure 5G) are not connected to their free substances, but form separate molecular families. Furthermore, in addition to the molecular families already mentioned above, there are a large number of tightly clustered nodes which were not annotated (Figure 5H) (Appendix A), thus highlighting the capability of molecular networks to reveal molecular diversity and potentially novel unidentified compounds, which requires further exploration and discovery.

The Inclusion of a metadata table is extremely valuable for interpreting the molecular networks generated using data. A Metadata File added to the analysis that describes the experimental setup and details allows for better downstream data visualization, analysis and interpretation [23]. As shown in Figure 5, the spectra of samples from PCA, PDA and PSA media are colored yellow, blue and pink, respectively. Metadata tables provide the necessary information to visualize the ‘origin’ of the detected metabolites, which facilitates a more intuitive and rapid visualization of the source of fungal metabolites and the amount of toxins produced. From the figure, the metabolites mainly came from PSA and PDA medium, and the metabolites produced from PCA were negligible, which also verified the previous conclusion that PCA medium had the worst toxigenic effect and PSA medium had the best toxigenic effect. With the help of metadata, it was found that *Alternaria* toxin has a distinct metabolic pathway on the PSA medium. In terms of toxin species, AOH, AME, ALU, ALT, DHA, DMDA, AOH-SO_3_, AME-SO_3_, DHA-SO_3_ and DMDA-SO_3_ are more suitable for culture with PSA media, while ATX-1, TEN and TeA are more suitable for culture with PDA, which provides us with a good preparation protocol for the study of individual toxins.

## 3. Materials and Methods

### 3.1. Analytical Standards

Altogether, eight certified standards of *Alternaria* toxins were obtained from Qingdao Pruibang Biological Engineering Co., Ltd. (Tsingtao, China), specifically alternariol (AOH), alternariol monomethyl ether (AME), altenuene (ALT), tentoxin (TEN), tenuazonic acid (TeA), altenusin (ALU), Altertoxin- I (ATX-I) and mycophenolic acid (MPA). The declared purity of all standards was in the range of 98–100.0%. The individual stock solutions of toxins were prepared at 1000 μg/mL in methanol. They were stored in amber vessels at −20 °C and brought to ambient temperature before use.

### 3.2. Chemicals and Materials

Acetonitrile, ethyl acetate, formic acid and methanol (all of LC-MS grade) were purchased from Fisher Scientific (Thermo Fisher Scientific, Waltham, MA, USA). Potato dextrose agar (PDA) medium was purchased from Beijing Land Bridge Technology Co., Ltd. (Beijing, China). The plate count agar (PCA) medium and potato sucrose agar (PSA) medium were prepared by our lab (preparation methods are shown in the Appendix A).

### 3.3. Sample Preparation and Extraction Methodology

Seven strains of *Alternaria* fungi were used in this study and were provided by the Laboratory of Mycotoxin and Health Research Center, Yantai University (Shandong, China) (Appendix A). For the investigation of *Alternaria* metabolites, strains were inoculated onto PCA, PDA and PSA medium for culturing. Strains were cultured as single-point inoculations onto mediums that were incubated at 25 °C in darkness for thirty days (Appendix A). Metabolites on cultured agar plates were extracted at the 10th, 20th and 30th days for the study of *Alternaria* metabolomic analysis.

Six plugs of each culture plate were extracted according to the method outlined by Smedsgaard [33]. Plugs of each culture strain were extracted with 4 mL of ethyl acetate. After vortexing for 3 min, the extract was sonicated at 35 °C for 30 min and subjected to water bath oscillation extraction for 30 min. The supernatant was collected and dried under nitrogen in a water bath at 45 °C before reconstitution in 1 mL of 3:1 methanol/water, filtered through 0.45 μm PTFE filters, and the filtrate was used for analysis.

### 3.4. HPLC-MS/MS Experiments

HPLC-MS/MS analysis of secondary metabolites was performed using a Q-Exactive Orbitrap mass spectrometer (Thermo Fisher Scientific, Waltham, MA, USA) with a heated electrospray ionization (HESI) source connected in tandem to an Agilent 1290 high-performance liquid chromatography (HPLC) system. Positive ionization modes were assessed for both HRMS and HRMS/MS. The separation was performed using a Hypersil GOLD C-18 column (100 mm × 2.1 mm, 1.9 μm). Chromatographic Separation was performed with a dual-solvent system with 0.1% (*v*/*v*) formic acid in water and acetonitrile (Appendix A). The injection was 5 μL and the flow rate was 0.3 mL/min. The column temperature was 40 °C.

Mass spectrometry conditions. Conditions were used for positive HESI: sheath gas, 45 units; auxiliary gas, 10 units; capillary voltage, 3.5 kV; capillary temperature, 320 °C; probe heater temperature, 350 °C; and S-Lens RF level, 45.0. All samples were analyzed by HRMS and HRMS/MS data-dependent acquisition (ddMS/MS) with the following settings: full scan resolution, 70,000; scan range, *m*/*z* 120–900; automatic gain control (AGC) target, 3.0 × 10^6^; maximum injection time (max IT), 100 ms; The TOP 5 intensity ions from the MS scan were sequentially mass selected under a 1.5 *m*/*z* isolation window and analyzed by HRMS/MS at a resolution of 17,500; AGC target, 1.0 × 10^5^; max IT 64 ms; collision energy parameters were: 20, 40, 70 eV; and threshold intensity, 1.0 × 10^3^.

### 3.5. HPLC-MS/MS Data Processing

The conjugated products were screened using the diagnostic fragmentation filtering (DFF) of MZmine 2. Refer to the experimental protocol of Li et al., the sulfated conjugated products diagnostic was investigated by setting neutral loss values at 79.9568 Da, the tolerance of *m*/*z* was 0.02Da or 5 ppm and base peak was set at 20%.

Thermo Raw HRMS files were converted into mzML files using MSConvert and imported to MZmine 2; the data were processed as per the method outlined by Li [26] with the following changes: the “chromatogram builder” was replaced with “ADAP (Automated Data Analysis Pipeline) chromatogram builder” module. New algorithms detected significantly fewer false positives in chromatographic peaks [34], which is advantageous to our data processing process (parameters are shown in the Appendix A).

### 3.6. Molecular Networking Parameters and Visualization

After the processing of the above steps, two data files were exported, which contained the feature quantification table (.csv file format) and the MS/MS spectral summary (mgf file format) for molecular networking in GNPS. Files were upload to a GNPS Advanced Analysis Tools module for feature-based molecular networking (FBMN) analysis with the following parameters: precursor ion mass tolerance, 0.02 Da; fragment ion mass tolerance, 0.02 Da; min pairs cosine score, 0.7; Network TopK, 10; minimum matched fragment ions, 6; maximum connected component size (Beta), 100; maximum shift between precursors, 500 Da. Additionally, the rest were default parameters. After the completion of the FBMN job, the output was downloaded as a (graphml) file and visualized in Cytoscape_3.8.2 [35]. In order to simplify the molecular network and easily find *Alternaria* mycotoxin clusters, some meaningless metabolic clusters composed of spectra were deleted from the molecular network. The data processing and analysis method is shown in Figure 6.

### 3.7. Identification of Alternaria Metabolites and Analysis

Xcalibur software was used to gather the information of retention time, exact masses of *m*/*z* precursor ions and characteristic fragment ions with *Alternaria* mycotoxins (AOH, AME, ALT, TEN, TeA, ALU, ATX-I and MPA) from eight existing commercial standards, which were used as confirmation standards for free metabolites (Appendix A). The confirmation standards of conjugated products were based on exact masses of *m*/*z* precursor ions, product ion fragments, neutral loss values and isotope peaks. Heatmaps for identified free and conjugated products were created using RStudio based on their response in mass spectrum. The GNPS platform was used to construct the molecular metabolic network, and the metabolic pathway of *Alternaria* mycotoxins was analyzed.

## 4. Conclusions

In this study, seven *Alternaria* fungi were cultured on three different media, and their metabolites were analyzed by ultra-high performance liquid chromatography tandem mass spectrometry to study their natural metabolites. The research results showed that the fungi cultured in PSA medium had the advantages of rapid colony growth rate, high toxigenic content and many toxigenic species, so it was more suitable for the cultivation of *Alternaria* fungi. HRMS/MS data sets were visibly analyzed and investigated by the combination of MZmine 2, Xcalibur, Cytoscape software and GNPS. By MZmine 2, sulfate conjugations were screened by mass spectrometry with an optimized DFF module based on a class-specific neutral loss of 79.9568 Da. These results were further confirmed with Xcalibur by comparison with unconjugated standards. The MS/MS spectral data of MZmine 2 were used for processing and imported into the GNPS-FBMN to establish a visual molecular networking to explore metabolic pathways of *Alternaria* mycotoxins. The results of FBMN showed that *Alternaria* mycotoxins had significant metabolic pathways with many types of metabolites. This research may provide theoretical support for the metabolomics study of *Alternaria*. At the same time, it is a promising strategy to analyze fungal metabolomics combined with a variety of visual data analysis tools. In future studies of fungal metabolomics, this strategy can also be applied to other types of fungi to improve the relevant fungal metabolism database.

## Figures and Tables

**Figure 1 molecules-28-03258-f001:**
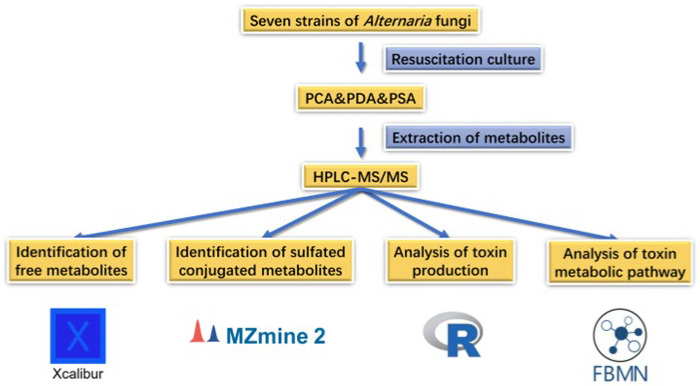
Research method.

**Figure 2 molecules-28-03258-f002:**
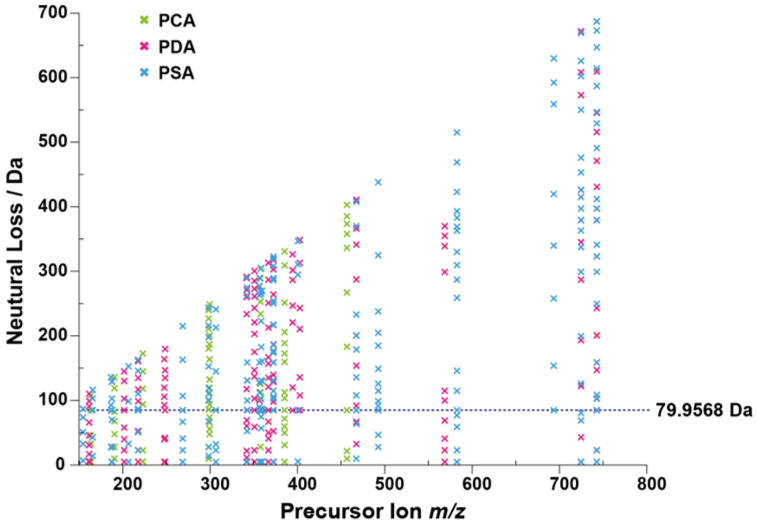
Neutral loss filtering of SO_3_ for sulfated products.

**Figure 3 molecules-28-03258-f003:**
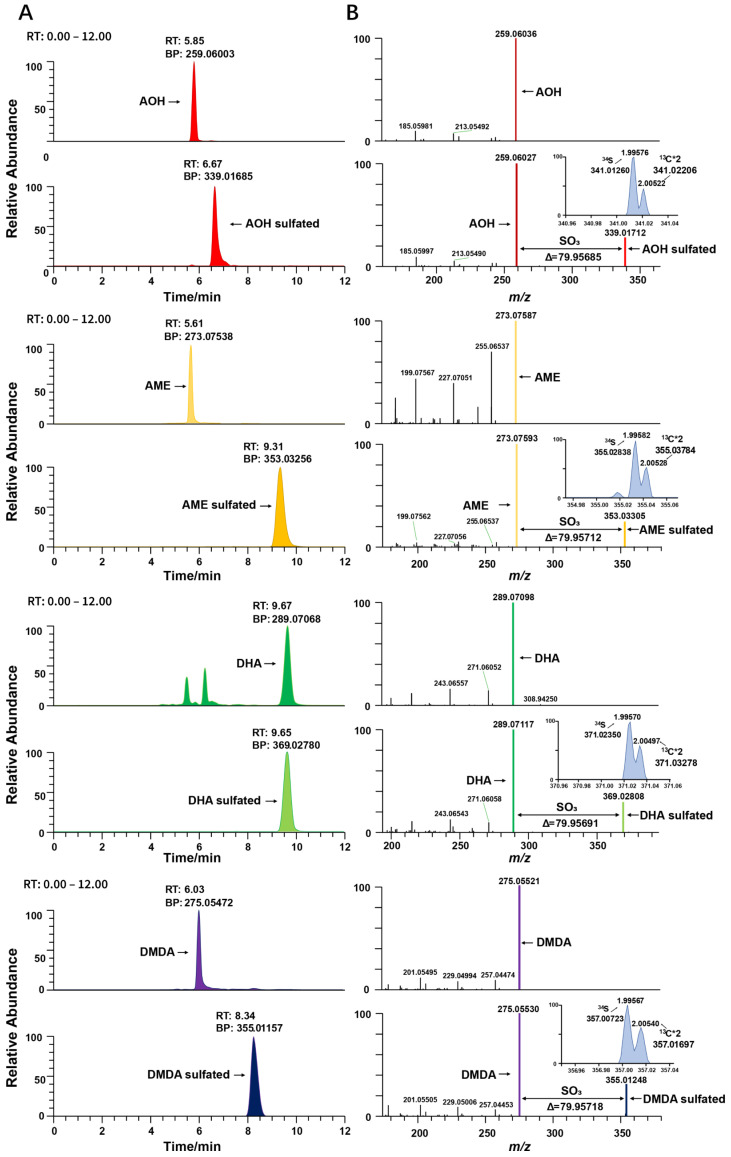
Example of detection of sulfate conjugated mycotoxin. (**A**) Chromatograms of sulfated and unsulfated mycotoxins. (**B**) Spectra of sulfated mycotoxins with neutral loss of SO_3_ (79.9568 Da ± 5 ppm) and isotope peak of ^34^S (1.995 Da).

**Figure 4 molecules-28-03258-f004:**
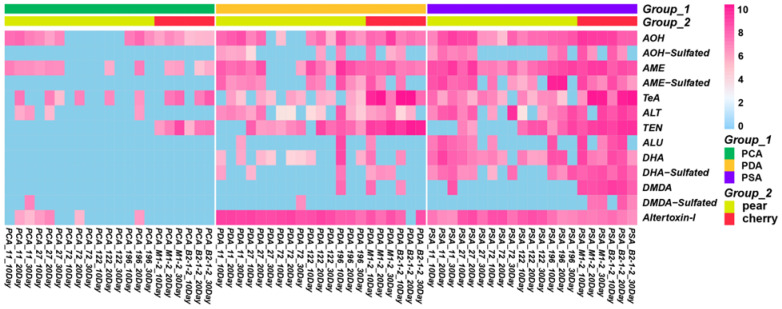
Heatmap for free *Alternaria* mycotoxins and sulfated mycotoxins by *Alternaria* fungi cultured in PCA, PDA and PSA medium.

**Figure 5 molecules-28-03258-f005:**
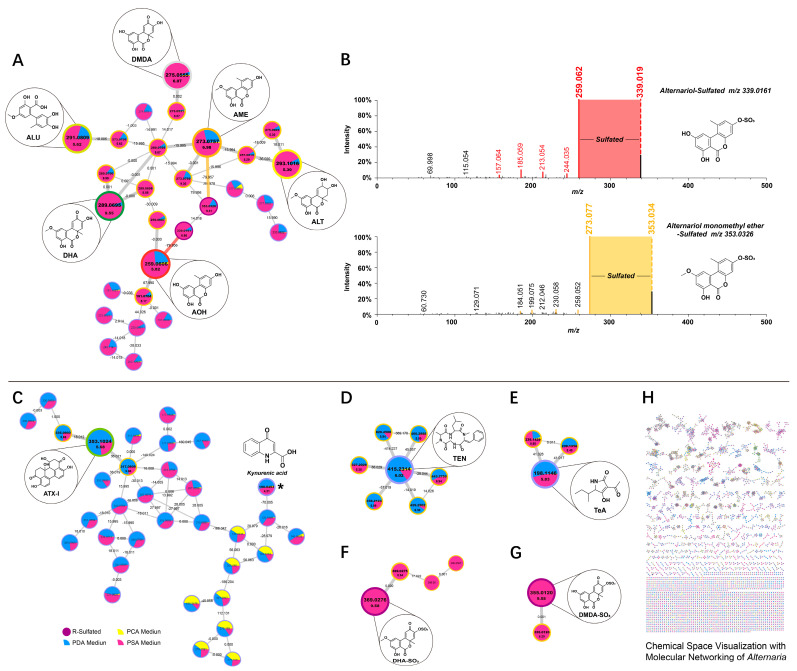
Visualization with Molecular Networking of the *Alternaria* natural product molecular family detected in PCA (yellow nodes), PDA (blue nodes) and PSA (pink nodes). (**A**,**C**–**G**) The molecular family of *Alternaria* metabolites. (**B**) Comparison of MS/MS spectra for AOH- and AME- sulfate conjugated. (**H**) Chemical space visualization with molecular networking of the *Alternaria*. (Appendix A).

**Figure 6 molecules-28-03258-f006:**
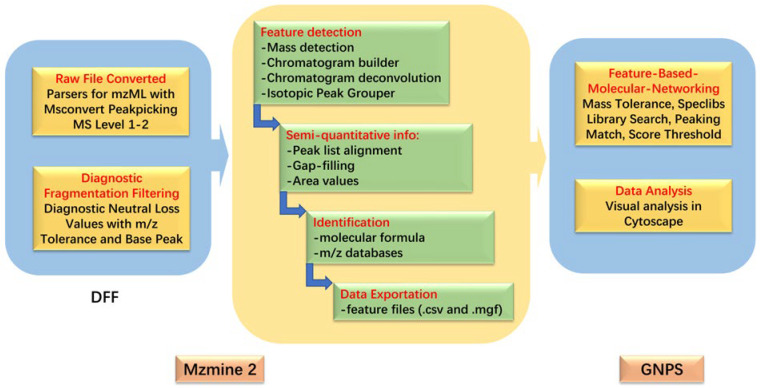
Data processing and analysis.

## Data Availability

Not applicable.

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
