# Peer review of "Metabolite Analysis of Alternaria Mycotoxins by LC-MS/MS and Multiple Tools"

_molecules, 2023, doi:10.3390/molecules28073258_

Round 1
Reviewer 1 Report
The manuscript presents an analysis of Alternaria mycotoxins metabolites with LC-HRMS supported by multiple tools for data analysis.
Studies of mycotoxins and their metabolites are essential for human health reasons. Each well-described method of fungi cultivation and their metabolites analyses plays a vital role in evaluating the influence of these dangerous organisms on our well-being. The authors stress that their work can extend knowledge of fungal metabolomics.
In the reviewed study, seven strains of Alternaria fungi were cultured on three mediums. For the chemical analysis, ultra-performance liquid chromatography with high-resolution mass spectrometry was used. The authors used eight analytical standards for qualitative and quantitative analysis of the metabolites. Obtained data were analyzed and visualized with multiple tools suitable for identifying free and sulphated conjugated metabolites, analysis of toxin production and their metabolic pathway. The description of the performed analysis is clear and consequent. Necessary figures and tables in the manuscript and supplementary materials support the discussion. However, some minor shortcomings are noticed after the revision article can be recommended for publication in Molecules MDPI Journal.
I would suggest the following revisions to be undertaken:
Abstract. The Authors use several abbreviations referring to analysis methods (HRMS/MS) and analyzed substances (TEN, TeA, AOH, AME, ALU, ALT, DHA and DMDA, ATX-1). Although it is clear when full text is explored, the abstract should contain full names of studied substances or used methods of analysis.
Abbreviations. The Authors use HRSM multiple times in the manuscript, but it is never connected with the full name of the method. Please include the clear meaning of the abbreviation.
Page 3, line 108. Please rephrase the sentence.
Page 10, lines 337-338. Please rephrase the sentence.
Figure 4. It needs to be clarified what represents the colours used in the bule-pink bar. The included scale needs to be extended with units and descriptions.
Figure 5. Please provide better resolution of included molecular networks. Figure 5H is not readable.
Author Response
Reviewer 1
The manuscript presents an analysis of Alternaria mycotoxins metabolites with LC-HRMS supported by multiple tools for data analysis.
Studies of mycotoxins and their metabolites are essential for human health reasons. Each well-described method of fungi cultivation and their metabolites analyses plays a vital role in evaluating the influence of these dangerous organisms on our well-being. The authors stress that their work can extend knowledge of fungal metabolomics.
In the reviewed study, seven strains of Alternaria fungi were cultured on three mediums. For the chemical analysis, ultra-performance liquid chromatography with high-resolution mass spectrometry was used. The authors used eight analytical standards for qualitative and quantitative analysis of the metabolites. Obtained data were analyzed and visualized with multiple tools suitable for identifying free and sulphated conjugated metabolites, analysis of toxin production and their metabolic pathway. The description of the performed analysis is clear and consequent. Necessary figures and tables in the manuscript and supplementary materials support the discussion. However, some minor shortcomings are noticed after the revision article can be recommended for publication in Molecules MDPI Journal.
Response: We thank the reviewer so much for the positive comment. We have revised the manuscript as suggested.
I would suggest the following revisions to be undertaken:
Abstract. The Authors use several abbreviations referring to analysis methods (HRMS/MS) and analyzed substances (TEN, TeA, AOH, AME, ALU, ALT, DHA and DMDA, ATX-1). Although it is clear when full text is explored, the abstract should contain full names of studied substances or used methods of analysis.
Response: As suggested, the full names have been added in the revised manuscript.
Abbreviations. The Authors use HRSM multiple times in the manuscript, but it is never connected with the full name of the method. Please include the clear meaning of the abbreviation.
Response: As suggested, the full name has been added in the revised manuscript.
Page 3, line 108. Please rephrase the sentence.
Response: As suggested, we have rephrased the sentence in the revised manuscript.
Page 10, lines 337-338. Please rephrase the sentence.
Response: As suggested, we have rephrased the sentence in the revised manuscript.
Figure 4. It needs to be clarified what represents the colours used in the bule-pink bar. The included scale needs to be extended with units and descriptions.
Response: As suggested, we have made corresponding revision in the revised manuscript.
Figure 5. Please provide better resolution of included molecular networks. Figure 5H is not readable.
Response: As suggested, we have improved Figure 5 in the revised manuscript.
Reviewer 2 Report
This study seems to very intresting and exploring about Metabolite Analysis of Alternaria Mycotoxins by LC-HRMS using bioinformatics tools.
However, before publication of this manuscript, I feel that the English should be improved, I suggest that the manuscript be edited by someone fluent in writing English (assuming that English is not the native language of the authors). A author will have to spend considerable time in revising the manuscript and quality of fighures too.
Author Response
Reviewer 2
This study seems to very interesting and exploring about Metabolite Analysis of Alternaria Mycotoxins by LC-HRMS using bioinformatics tools.
Response: We thank the reviewer so much for the positive comment.
However, before publication of this manuscript, I feel that the English should be improved, I suggest that the manuscript be edited by someone fluent in writing English (assuming that English is not the native language of the authors). An author will have to spend considerable time in revising the manuscript and quality of figures too.
Response: We thank the reviewer for the suggestion. As suggested, a native English-speaking colleague has improved the manuscript. In addition, Figures also have been improved in the revised manuscript.
Reviewer 3 Report
The authors try to describe the identification of metabolites of mycotoxins generated by Alternaria. Unfortunately, there are many sentences which are grammatically wrong and/or make little sense. That makes reading the manuscript very difficult. The authors also constantly use abbreviation without prior definitions (abstract and elsewhere). The term "tandem HRMS/MS" is wrong. It is either tandem-MS or MS/MS with the later being the preferable term.
It is unclear what Fig. 2 is supposed to show. Neither the text nor the legend help in interpreting it.
Fig. 3 and the text related to it are incorrect. The isotopic peak at +m/z 2.005 is not coming from 2*13C which has a probability of appearing in a C20 compound of only 0.2 %. It stems from 18O which, given the number of O in, f.i., sulfated AME of 8, has a probability of appearing of 1.6 %. With the LC conditions used by the authors the retention of sulfated compounds should be less than the retention of the respective parent compound and the detectability in positive ESI mode be much worse. All of the above is contradicted by Fig. 3.
In the supplemental material the authors hint at changes in chromatography when changing the detection mode from negative to positive ESI. Only changing the separation setup would do this but that is nowhere mentioned.
This casts a lot of doubt on the findings in the rest of the manuscript.
Author Response
Reviewer 3
The authors try to describe the identification of metabolites of mycotoxins generated by Alternaria. Unfortunately, there are many sentences which are grammatically wrong and/or make little sense. That makes reading the manuscript very difficult. The authors also constantly use abbreviation without prior definitions (abstract and elsewhere). The term "tandem HRMS/MS" is wrong. It is either tandem-MS or MS/MS with the later being the preferable term.
Response: Thank you very much for your kind comments. As suggested, the English language has been revised through the whole work. The full names of abbreviation also have been defined when they first appeared. That mistake “tandem HRMS/MS” has been corrected in the revised manuscript.
It is unclear what Fig. 2 is supposed to show. Neither the text nor the legend help in interpreting it.
Response: A detailed explanation on Figure 2 on page 3 has been provided in the revised manuscript.
“The m/z was set to 79.9568 for the diagnostic production. Processing the data to gener-ate the neutral loss plot of Alternaria (Figure 2), the neutral loss plot could visually ex-tract showing precursor metabolite m/z, and neutral mass losses. Taking the extraction results of strain NO.196 as an example, a total of 6 compounds were detected with a neutral loss of 79.9568 (±5ppm) in medium, including 1 for PCA, 2 for PDA, and 4 for PSA plates (Supplementary material Table S4).”
Fig. 3 and the text related to it are incorrect. The isotopic peak at +m/z 2.005 is not coming from 2*13C which has a probability of appearing in a C20 compound of only 0.2 %. It stems from 18O which, given the number of O in, f.i., sulfated AME of 8, has a probability of appearing of 1.6 %. With the LC conditions used by the authors the retention of sulfated compounds should be less than the retention of the respective parent compound and the detectability in positive ESI mode be much worse. All of the above is contradicted by Fig. 3.
Response: Thank the reviewer for the valuable comments. We have revised the description on Fig. 3 in the manuscript.
In nature, there are two sulfur elements with exact mass of 31.97207, and 33.96787. The mass shift is 1.9958 for one sulfur isotope. From figure 3, the four conjugated compounds exhibit sulfur isotope peaks of 341.01260, 355.02838, 371.02350, and 357.00723 with mass shift of 1.995 isotope peaks. The results indicate the four conjugated compounds are sulfated Alternaria mycotoxins.
The same as sulfur element, there are two carbon elements with exact mass of 13.00335 and 12.00000. The mass shift is 1.00335 for one carbon elements. In figure 3, mass shift of 2.005 is observed, and it is speculated that the mass shift might be from two 13 C isotope peaks. Figure 3 keeps correspondence to the previous literature “Rapid Commun. Mass Spectrom. 2015, 29, 1805–1810 DOI: 10.1002/rcm.7286”
As the reviewer mentioned, Alternaria mycotoxins were reported to show high response in ESI negative mode. And the retention time of conjugated Alternaria mycotoxins were earlier than the precursor in ESI negative mode. However, in our work, the most optimized ionization mode of these targets was observed in ESI positive mode with 0.1% Formic acid and acetonitrile as mobile phase. And in the ESI+ mode with acid mobile phase, retention times of sulfated mycotoxins are later than the precursors, which is different from basic solution as mobile phase.
In the supplemental material the authors hint at changes in chromatography when changing the detection mode from negative to positive ESI. Only changing the separation setup would do this but that is nowhere mentioned. This casts a lot of doubt on the findings in the rest of the manuscript.
Response: Thank the reviewer for the comment. The chromatograms in the positive and negative modes have been provided in the supplemental material for better understanding.
Thanks for the reviewer’s kind comments. We hope our work is much more acceptable now. If there are still some other problems, we will be glad to make any revisions to make our work acceptable by the journal.